# Antioxidant Profile of *Origanum dictamnus* L. Exhibits Antiaging Properties against UVA Irradiation

Sophia Letsiou [1,2,*], Maria Trapali [1], Despina Vougiouklaki [2], Aliki Tsakni [2], Dionysis Antonopoulos [2] and Dimitra Houhoula [2]

[1] Department of Biomedical Science, University of West Attica, Agiou Spyridonos 28, 12243 Egaleo, Greece; ymaria@uniwa.gr

[2] Department of Food Science and Technology, Faculty of Food Sciences, University of West Attica, 12243 Egaleo, Greece; dvougiouklaki@uniwa.gr (D.V.); atsakni@uniwa.gr (A.T.); antondion@uniwa.gr (D.A.); dhouhoula@uniwa.gr (D.H.)

* Correspondence: sletsiou@uniwa.gr or sletsiou@gmail.com

**Abstract:** Skin aging mainly occurs due to intrinsic and extrinsic factors. Extrinsic aging is a consequence of exposure to ultraviolet radiation. Meanwhile, natural products exhibit protective properties against skin aging as well as photoaging. In this context, the research on natural anti-aging agents is greatly advanced, and in recent years, numerous plant-based products have been investigated. The aim of this study was to assess the antioxidant profile of *Origanum dictamnus* L. extract as well as its antiaging effects on 2D cultures of fibroblasts and keratinocytes under UVA irradiation to unravel the potential role of *Origanum dictamnus* L. in cosmetology. In an attempt to explore the antioxidant profile of the extract, we employed well-established enzymatic assays (DPPH, FRAP, ABTS, and TPC) and a phytochemical screening by LC/MS. According to our findings, the *Origanum dictamnus* L. extract possesses high scavenging activity (DPPH, ABTS), high phenolic content (TPC), and high Fe(III)-reduction activity (FRAP). Moreover, the LC/MS analysis revealed that the extract was rich in flavonoids, holding a high content of curcumin, kampferol, silymarin, cyanidin-3-glucoside, deosmin, rutin, and quercetin. To gain insight into the bioactivity of *Origanum dictamnus* L. extract in cell aging, the expression of various genes that are implicated in the skin aging process in keratinocytes and fibroblasts was studied. The gene expression analysis revealed that the extract increases cell proliferation in the cells exposed to UVA irradiation and concomitantly modulates the expression of genes related to the aging process in keratinocytes (KLK7, OCLN, GBA1) and fibroblasts (SIRT2, FOXO3, COL3A1) under the same conditions.

**Keywords:** *Origanum dictamnus* L.; natural extract; fibroblasts; photoaging

## 1. Introduction

Aging is a physiological process that every living creature experiences. In fact, aging is a major factor in a number of degenerative diseases such as cancer, Alzheimer's disease, and cardiovascular disease (CVD), which are chronic diseases that cause fatalities in humans [1]. It has been reported that every day 150,000 people pass away due to aging, and approximately two-thirds pass away due to age-related diseases [2]. Furthermore, at a molecular level, both keratinocytes and fibroblasts—the main cellular components of the epidermis and dermis layers, respectively—are affected by the skin aging process depending on intrinsic or extrinsic factors. Both keratinocytes and fibroblasts lose their ability to renew and communicate with essential molecules for skin barrier function [3]. In addition, it has been recently reported that the physical interaction between keratinocytes and fibroblasts is impaired due to aging processes [4].

It is believed that free radicals such as reactive nitrogen species (RNS) and reactive oxygen species (ROS) have a rather significant role in age-linked functional alterations [5,6]. Indeed, it has been suggested that the primary cause of aging is the oxidative stress that

occurs in mitochondria as a consequence of cellular respiration [7]. Normally, free radicals are produced in our organism as the outcome of ATP (adenosine triphosphate) production in the mitochondria when cells use oxygen (an essential element for life) to produce energy [8]. Thus, an imbalance between the body's excess production or accumulation of free radicals and its capacity to detoxify the reactive chemicals leads to oxidative stress, which is a major contributor to the onset of a number of degenerative and age-related chronic diseases [9,10]. In addition, the overproduction of ROS in the skin stimulates the expression of metalloproteinases (MMPs) [11], resulting in the degradation of collagen, which has a fundamental role in extracellular matrix (ECM) organization and skin function [12].

Therefore, the delay or inhibition of aging-related disorders using natural products is quite an attractive strategy to counteract aging and promote a healthy way of living [2]. Antioxidants are natural compounds that are present in fruits and vegetables and protect the cell from free radical damage through radical neutralization and scavenging. In fact, several research studies have reported that antioxidant intake is correlated with a lower prevalence of chronic diseases [13–16].

Specifically, extracts from flowers and herbs are becoming quite popular and are incorporated into different products as they contain a variety of essential molecules with anti-aging, antioxidant, and anti-inflammatory properties [12,17]. *Origanum dictamnus* L., or Cretan dittany, is a perennial and aromatic plant that only grows wild in the mountainsides and gorges of the island of Crete in Greece [18]. It has been reported that *Origanum dictamnus* L. extracts exhibited several biological properties such as antioxidant [19–22], antimicrobial (against bacteria, protozoans, and fungi [23–27] and cytotoxic activity against P388 (murineleukemia) and human bronchial epidermoid lung cancer NSCLC-N6 cell lines, respectively [28]. Moreover, in 2012, a monograph for *Origanum dictamnus* L. was approved as a traditional herbal medicinal product either for oral or cutaneous use (as infusion or decoction) among EU countries [29,30]. Nevertheless, little is known to date regarding its potential role in skin aging.

In this study, we sought to explore the antioxidant content of *Origanum dictamnus* L. extract (OdLe) and its potential impact on the cell aging process under UVA irradiation, focusing on human keratinocytes and fibroblasts.

## 2. Materials and Methods

### 2.1. Chemicals

Extract Preparation

The extraction preparation was based on a previously reported method [31]. Briefly, 100 mg of dried and powdered leaves of *Origanum dictamnus* L. from the region of Crete were extracted with 1 mL of extraction solvent (80% pure methanol, 20% water) for 10 min in an ultrasonic bath at a temperature of 25–30 °C. This was followed by centrifugation for 5 min at 2500 rpm. The species name is according to Dimopoulos et al. [32]. The extraction procedure was carried out in triplicate.

### 2.2. Determination of Antioxidant Activity

2.2.1. Determination of Total Phenolic Content (TPC)

The determination of the TPC of the extracts was performed as previously [33] using the Folin Ciocalteu method. Specifically, in a glass tube, 100 µL of extract was mixed with 500 µL of water and 100 µL of Folin-Ciocalteu reagent. Each reaction mixture was allowed to stand for 6 min, followed by the addition of 1 mL of 7% sodium carbonate and 500 µL of distilled water. Absorbance at 760 nm was recorded at 37 °C in a water bath using a Shimazu UV-VIS spectrophotometer. A standard curve of gallic acid (0.05–0.4 mg/mL) was constructed. All determinations were carried out in triplicate, and the results are expressed as mg gallic acid equivalent (GAE)/g of extract.

### 2.2.2. Ferric Reducing Antioxidant Potential (FRAP) Assay

The FRAP assay was performed according to a previously reported and well-known method [34]. This method is based on the reduction, at low pH, of a colorless ferric complex ($Fe^{3+}$-tripyridyltriazine) to a blue-colored ferrous complex ($Fe^{2+}$-tripyridyltriazine) by the action of electron-donating antioxidants. The reduction is monitored by measuring the change in absorbance at 593 nm. The working FRAP reagent was prepared daily by mixing 10 volumes of 300 mM acetate buffer, pH 3.6, with 1 volume of 10 mM TPTZ (2,4,6-tri(2-pyridyl)-s-triazine) in 40 mM hydrochloric acid and with 1 volume of 20 mM ferric chloride. A standard curve was prepared using various concentrations of $FeSO_4 \times 7H_2O$. All solutions were used on the day of preparation. One hundred microliters of sample solutions and 300 µL of deionized water were added to 3 mL of freshly prepared FRAP reagent. The reaction mixture was incubated for 5 min at 37 °C in a water bath. Then, the absorbance of the samples was measured at 593 nm. A sample blank reading using an acetate buffer was also taken. The difference between sample absorbance and blank absorbance was used to calculate the FRAP value. In this assay, the reducing capacity of the plant extracts tested was calculated with reference to the reaction signal given by a $Fe^{2+}$ solution. FRAP values were expressed as µM $Fe^{2+}$ of the sample. All measurements were performed in triplicate. Gallic acid (0.24–3.84 µg/mL) was used as the positive control.

### 2.2.3. Free Radical Scavenging by the Use of the DPPH Radical

The DPPH assay was performed according to a previously modified method [35,36]. DPPH radicals have an absorption maximum of 515 nm, which disappears with reduction by an antioxidant compound. The DPPH• solution in methanol ($6 \times 10^{-5}$ M) was prepared daily, and 3 mL of this solution was mixed with 100 µL of methanolic solutions of plant extracts. The samples were incubated for 20 min at 37 °C in a water bath, and then the decrease in absorbance at 515 nm was measured (AE). A blank sample containing 100 µL of methanol in the DPPH• solution was prepared daily, and its absorbance was measured (AB). The experiment was carried out in triplicate. Radical scavenging activity was calculated using the following formula: % inhibition = [(AB − AE)/AB] × 100, where AB = absorbance of the blank sample and AE = absorbance of the plant extract. Gallic acid (1.30–333.33 µg/mL) was used as the positive control.

### 2.2.4. Free Radical Scavenging by the Use of the ABTS Radical

The determination of the free radical scavenging capacity of plant extracts was performed according to a previously reported method [37]. The method is based on the reduction of ABTS+• radicals by antioxidants in the plant extracts tested. ABTS was dissolved in deionized water to a 7 mM concentration. ABTS radical cation (ABTS+•) was produced by reacting ABTS solution with 2.45 mM potassium persulfate (final concentration) and allowing the mixture to stand in the dark at room temperature for 12–16 h before use. For the study, the ABTS+• solution was diluted in deionized water or ethanol to an absorbance of 0.7 (±0.02) at 734 nm. An appropriate solvent blank reading was taken (AB). After the addition of 100 µL of aqueous or ethanolic (according to solubility) plant extract solutions to 3 mL of ABTS+• solution, the absorbance reading was taken at 25 °C 10 min after initial mixing (AE). All solutions were used on the day of preparation, and all determinations were carried out in triplicate. The percentage of inhibition of ABTS+• was calculated using the above formula: % inhibition = [(AB − AE)/AB] × 100, where AB = absorbance of the blank sample and AE = absorbance of the plant extract. Gallic acid (0.39–100 µg/mL) was used as the positive control.

### 2.2.5. LC-(ESI)/MS Analysis for Plant Extract

The chemical determination of the plant extract was based on LC-ESI single quadrupole MS (Shimadzu, Kyoto, Japan) using a recently published method [38] to assess the profile of the antioxidant metabolites in *Origanum dictamnus* L. extract. Briefly, the LC was performed on the reversed-phase SyncronisTM C-18 column 150 × 4.6 mm; particle size: 5 µm (Thermo

Fisher Scientific^TM, Waltham, MA, USA). with a flow rate of 0.8 mL/min. The system was operated in gradient mode at a temperature of 45 °C. The mobile phase consisted of 50% acetonitrile (ACN) with 50% ddH$_2$O (solvent A) and 95% dH$_2$O, 5% MeOH, and 0.2% acetic acid (solvent B). A linear gradient according to 0–20 min 20% B, 20–40 min 55% B, 40–55 min 100% B, 55–65 min 100% B, 66–75 min 10% B. The identification was based on the retention times as well as overlay curves. Phenolic compounds were identified by comparison with the retention time of the standards of phenolic compounds [38]. The results were expressed as the mean value ± SD of three measurements.

### 2.3. Assessments of Antiaging Activity

### 2.3.1. 2D Human Cell Cultures

The 2D human cells (Lonza, Walkersville, MD, USA) were cultured as described previously [39,40].

### 2.3.2. UVA Irradiation Treatment

Following incubation with the extract (0.5 µg/mL) for 48 h, the cells were exposed to 10 J/cm$^2$ of UVA light (lamp equipped with a 400 w ozone-free Philips HPA lamp, UV type 3) between 300 and 400 nm at a distance of 20 cm from the cell cultures for 30 min [41,42]. Prior to exposure to UVA light, the cells were washed twice with phosphate-buffered saline (PBS).

### 2.3.3. Cell Viability AssessmentsATP Determination

ATP Determination

Intracellular ATP levels were determined using the ATP determination kit (Thermo Fisher Scientific) as described before [43,44]. Briefly, NHDF cells were incubated for 48 h with different concentrations of OdLe (10 µg/mL, 5 µg/mL, 1 µg/mL, 0.5 µg/mL) before the ATP determination. The intracellular levels of ATP were determined under or without UVA irradiation. Three independent experiments, along with eight technical repetitions in each experiment, were carried out to validate the whole process.

RNA Isolation, Reverse Transcription-PCR, and Real-Time PCR

The RT-qPCR reactions were run in a Real-Time PCR System (Applied Biosystems, Foster City, CA, USA) using SYBR Select Master Mix (Applied Biosystems, Foster City, CA, USA), gene-specific primers at a final concentration of 0.5 µM each, and 500 ng of total RNA for cDNA synthesis. Total RNA was isolated and purified using the Nucleospin RNA kit (Macherey-Nagel, Düren, Germany). The relative gene expression was based on the comparative Ct ($2^{-\Delta\Delta Ct}$) method, while the normalization was based on two reference genes (ACTB and GAPDH [45,46]. Supplemental Table S1 shows the relative expressions of the samples, while primers are presented in Supplemental Table S2.

Moreover, in this study, we examined 8 different experimental states: untreated NHEK cells under or without UVA irradiation; untreated NHDF cells under or without UVA irradiation; NHEK cells treated with OdLe (0.5 µg/mL) under or without UVA irradiation; and NHDF cells treated with OdLe (0.5 µg/mL) under or without UVA irradiation. The whole analysis was based on three biological replicates and eight technical repeats.

### 2.3.4. Statistical Analysis

Statistical calculations are based on the mean of separate trials (each individual experiment with three biological replicates and eight technical repeats) using the program GraphPad Prism version 5 (GraphPad Software Inc., San Diego, CA, USA). Statistical significance was calculated using an analysis of variance (ANOVA) with Bonferroni multiple comparisons. A *p*-value below 0.05 was regarded as significant and is depicted with an *.

## 3. Results

### 3.1. Antioxidant Capacity

Four different enzymatic approaches, namely DPPH, TPC, ABTS, and FRAP, were utilized in order to assess the antioxidant capacity of the *Origanum dictamnus* L. extract.

Free Radical Scavenging by the Use of the DPPH Radical

In the DPPH assay, GA was used as a standard, which exhibited the highest DPPH radical scavenging activity with an $IC_{50}$ value of $18.69 \pm 0.1$ mg/mL. The OdLe exhibited significant DPPH radical scavenging activity with an $IC_{50}$ value of $22.19 \pm 0.1$ mg/mL yet lower antioxidant activity compared to GA.

Determination of Total Phenolic Content (TPC)

The total phenolic content was assessed using a Folin-Ciocalteu assay. According to this assay, the total phenolic content of the OdLe was $200 \pm 0.001$ mg/g crude extract.

Free Radical Scavenging by the Use of the ABTS Radical

Moreover, the OdLe showed a significant ability to inhibit $98.8 \pm 0.1\%$ of the ABTS radical, which was quite similar to the GA ability ($100 \pm 0.1\%$).

Ferric Reducing Antioxidant Potential (FRAP) Assay

According to the FRAP assay, the antioxidant capacity of the OdLe was $2688 \pm 38.19$ μM $FeSO_4 \cdot 7H2O$ slightly lower compared to the antioxidant ability of GA ($3688 \pm 35.19$ μM $FeSO_4 \cdot 7H2O$).

### 3.2. Chemical Characterization with LC/MS

An LC-ESI-MS approach was used to reveal the phytochemical profile of the OdLe so as to evaluate different types of polyphenols such as flavonoids, phenolic acids, and anthocyanins. According to the LC-ESI/MS analysis, the OdLe was rich in flavonoids. In addition, certain phenolic acids were detected in abundance. Specifically, the most dominant polyphenolic compounds in the OdLe were: curcumin, kampferol, silymarin, cyanidin-3-glucoside, deosmin, rutin, tartric acid, coumaric acid, and quercetin (Table 1).

**Table 1.** Detection of polyphenols in the *Origanum dictamnus* L. extract (50 ppb).

| | Identified Molecule | Ionization Mode | *m/z* | Molecular Formula | Plant Extract 50 ppb |
|---|---|---|---|---|---|
| 1 | Pyrogallic acid | (−) | 125 | $C_6H_6O_3$ | 5.81 |
| 2 | Na-salicylate | (−) | 109 | $C_6H_6O_2$ | 18.24 |
| 3 | Tartaric acid | (−) | 104 | $C_4H_6O_6$ | 31.83 |
| 4 | Citric acid | (−) | 191 | $C_{10}H_{12}O_5$ | 2,53 |
| 5 | Quercetin | (−) | 300.5 | $C_{15}H_{10}O_7$ | 23.31 |
| 6 | Rutin | (−) | 609 | $C_{27}H_{30}O_{16}$ | 32,51 |
| 7 | Curcumin | (−) | 366.80 | $C_{21}H_{20}O_6$ | 64.15 |
| 8 | Catechin | (−) | 289 | $C_{15}H_{14}O_6$ | 12.33 |
| 9 | Silymarin | (−) | 481 | $C_{25}H_{22}O_{10}$ | 48.72 |
| 10 | Deosmin | (−) | 607 | $C_{28}H_{32}O_{15}$ | 38.15 |
| 11 | Kampferol | (−) | 285 | $C_{15}H_{10}O_6$ | 51.08 |
| 12 | Coumaric acid | (−) | 163 | $C_9H_8O_3$ | 24.45 |
| 13 | Cinnamic acid | (−) | 149.06 | $C_9H_8O_2$ | 2.35 |

**Table 1.** *Cont.*

| Identified Molecule | | Ionization Mode | *m/z* | Molecular Formula | Plant Extract 50 ppb |
|---|---|---|---|---|---|
| 14 | Gallic acid | (−) | 169.17 | $C_7H_6O_5$ | 4.46 |
| 15 | Ascorbic acid | (−) | 175 | $C_6H_8O_6$ | 2.35 |
| 16 | Cyanidin-3 glucoside | (+) | 484 | $C_{21}H_{21}ClO_{11}$ | 48.26 |

*3.3. Assessments of Antiaging Activity*

To evaluate the antiaging activity of the OdLe, we started with cell viability and cytotoxicity assessments in NHDF cells. We targeted the intracellular levels of ATP by examining different concentrations of OdLe (0.5 μg/mL). Figure 1a,b represent the cell viability assessments in NHDF cells under or without UVA irradiation. Our findings demonstrated that the OdLe enhanced cell viability in NHDF cells in all the tested concentrations ($p < 0.05$) compared to the control (untreated NHDF). In addition, we explored cell viability under UVA irradiation after the cells were treated with the lowest concentration of OdLe (Figure 1b). According to the analysis, the OdLe was not only proven non-cytotoxic under UVA irradiation but also maintained intact the initial levels of cell viability in NHDF cells ($p < 0.05$, Figure 1b).

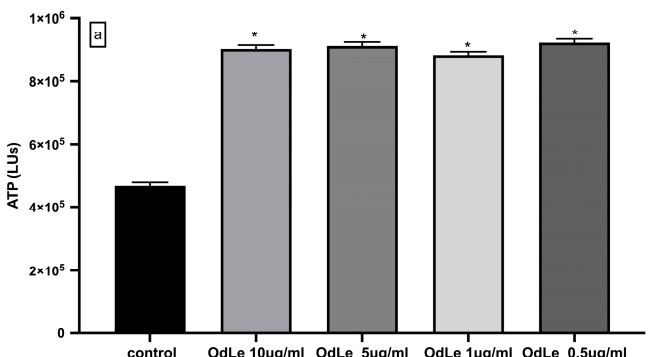

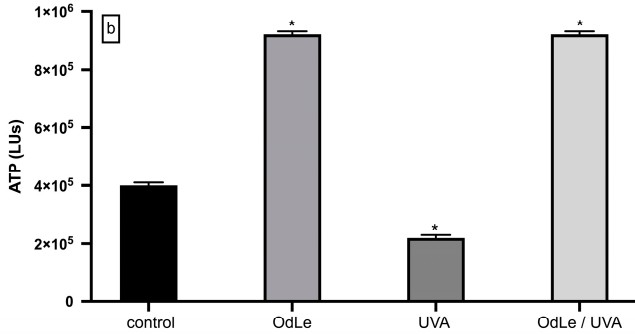

**Figure 1.** (**a**) ATP intracellular levels (LUs) indicated as mean ± SEM for control and NHDF cells treated with OdLe (0.5 μg/mL). * $p < 0.05$ significant difference from the control (ANOVA test). (**b**) ATP intracellular levels (LUs) indicated as mean ± SEM for Control: untreated NHDF cells, OdLe: NHDF cells treated with 0.5 μg/mL OdLe, UVA: NHDF under UVA irradiation, and OdLe/UVA: NHDF treated with 0.5 μg/mL OdLe under UVA irradiation. * $p < 0.05$ significant difference from the control (ANOVA test).

To gain insight into the antiaging effect and the related molecular mechanisms of OdLe bioactivity in human keratinocytes and fibroblasts under UVA irradiation, the expression of an array of genes involved in the aging process was studied. Figure 2 displays the

relative transcript levels that were significantly changed with the addition of OdLe in NHEK cells. As displayed in Figure 2, the genes occludin (OCLN), inhibin subunit beta A (INHBA), UDP-glucose ceramide glucosyltransferase (UGCG), b-glycocerebrosidase (GBA1), kallikrein-related peptidase 7 (KLK7), and corneodesmosin (CDSN) were up-regulated in NHEK cells treated with OdLe (0.5 μg/mL) under UVA irradiation compared to control and other conditions ($p < 0.05$). In addition, the transcripts of the gene OCLN were up-regulated in NHEK cells treated with OdLe (0.5 μg/mL) as compared to control and other conditions ($p < 0.05$).

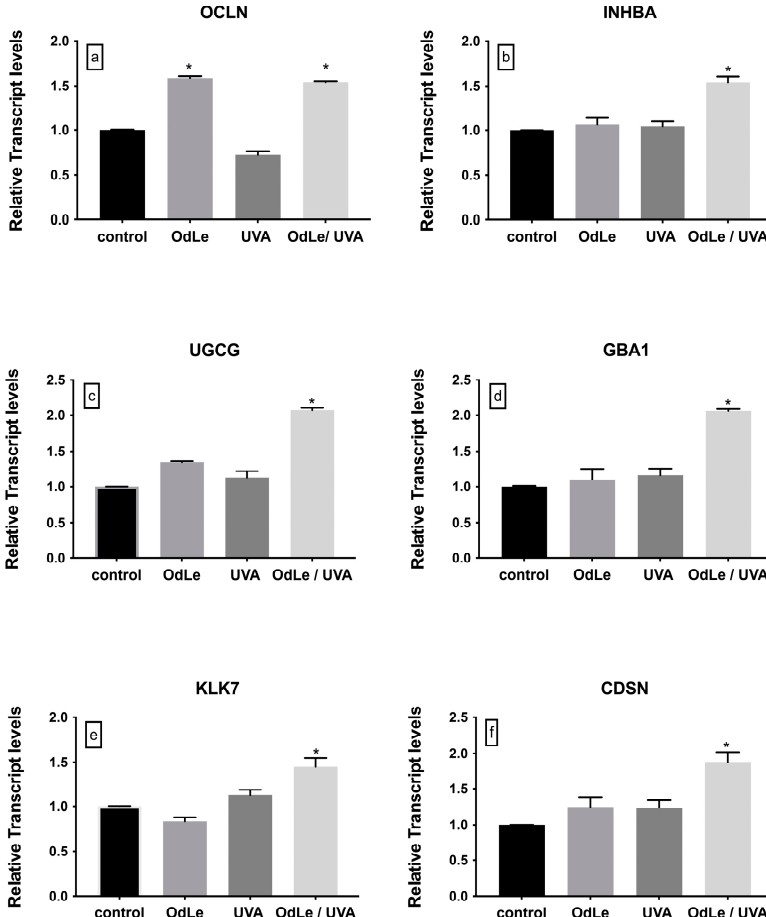

**Figure 2.** Gene expression levels of: OCLN (**a**), INHBA (**b**), UGCG (**c**), GBA1 (**d**), KLK7 (**e**), and CDSN (**f**) in NHEK cells, indicated as a fold change ± SEM. Experimental conditions: control: untreated NHEK cells, OdLe: NHEK cells treated with OdLe (0.5 μg/mL), UVA: NHEK cells under UVA irradiation, OdLe/UVA: NHEK cells treated with OdLe (0.5 μg/mL) under UVA irradiation. For the transcriptomic analysis ACTB and GADPH were used as internal reference genes. * $p < 0.05$ indicates groups significantly different from the control (ANOVA test).

Furthermore, Figure 3 depicts the results from the gene expression analysis in NHDF cells targeting the cell aging process. Specifically, the transcripts of the genes *sirtuin1 (SIRT1)*, *sirtuin 2 (SIRT2)*, *forkhead box O3 (FOXO3)*, *CD44 molecule (CD44)*, *collagen type III alpha 1 chain (COL3A1)*, and *matrix metallopeptidase 14 (MMP14)* were upregulated in NHDF cells treated with OdLe (0.5 μg/mL) under UVA irradiation as compared to control and other conditions ($p < 0.05$). Moreover, the transcripts of the genes *FOXO3* and *COL3A1* were upregulated in NHDF cells only with the addition of OdLe (0.5 μg/mL) as compared to control and other conditions ($p < 0.05$).

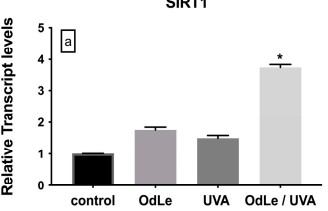
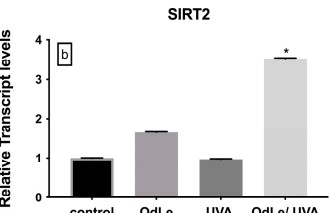
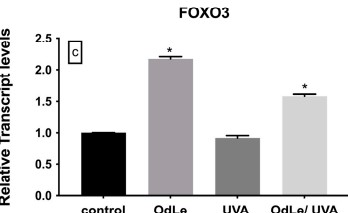
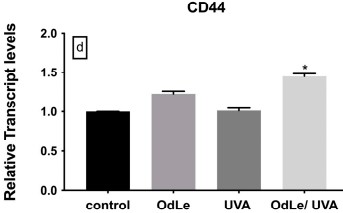
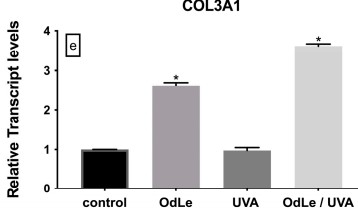
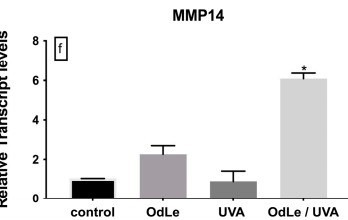

**Figure 3.** Gene expression levels of SIRT1 (**a**), SIRT2 (**b**), FOXO3 (**c**), CD44 (**d**), COL3A1 (**e**), MMP14 (**f**) in NHDF cells, indicated as a fold change ± SEM. Experimental conditions: control: untreated NHDF cells, OdLe: NHDF cells treated with OdLe (0.5 μg/mL), UVA: NHEK cells under UVA irradiation, OdLe/UVA: NHDF cells treated with OdLe (0.5 μg/mL) under UVA irradiation. For the transcriptomic analysis ACTB and GADPH were used as internal reference genes. * $p < 0.05$ indicates groups significantly different from the control (ANOVA test).

## 4. Discussion

Plant extracts are considered to have beneficial properties for skin health [47,48]. For instance, it has been reported that plant extracts have anti-aging properties, mainly attributed to their antioxidant properties against UVA irradiation (the photoaging process), which is considered the prime stimulator of skin aging [47–49]. Moreover, plant polyphenolic compounds are suggested to be the major contributors to the antioxidant capacity of plants, which can attenuate photoaging by scavenging reactive oxidants [47,49]. The relationship between phenolic compound content and antioxidant function has attracted considerable attention, and many researchers have confirmed a linear correlation between the two [49]. Thus, in this study, we explored the antioxidant profile of *Origanum dictamnus* L. extract as well as its impact on fibroblasts under UVA irradiation in an attempt to unravel the potential role of *Origanum dictamnus* L. in cosmetology as well as dermocosmetics.

Initially, we evaluated the antioxidant content of OdLe using four representative in vitro assays, namely FRAP, DPPH, TPC, and ABTS. According to our results based on the DPPH assay, the OdLe possesses strong antioxidant activity, as it has been reported that extracts that possess IC50 values ranging from 50 to 100 mg/mL are considered to exhibit intermediate antioxidant activity, while extracts with IC50 values ranging between 10 and 50 mg/mL are considered to possess strong antioxidant activity [50]. In addition, our findings in the ABTS assay are in agreement with previous reports [51–54] corroborating the major scavenging activity of the extract. Moreover, the total phenolic content of the OdLe was determined, since phenolics constitute one of the major groups of bioactive plant

compounds that act as primary antioxidants or free radical terminators. Our results have shown a TPC of 200 mg GA/g of the OdLe, which agrees with another recent study [55]. Furthermore, our findings based on the FRAP assay demonstrated that OdLe possesses a Fe(III)-reduction-related activity comparable to GA-related activity. Fe(III)-reduction is often used as an indicator of electron-donating activity, which is an important mechanism of phenolic antioxidant action [53,56,57].

In an attempt to explore more of the antioxidant profile of the OdLe, we proceed with the exploitation of its phytochemical composition, targeting its phenol content. For this purpose, an LC-MS platform, which is extensively used to explore metabolite profiling, was employed. Based on the phytochemical screening by LC/MS, the OdLe was particularly rich in flavonoids but also in phenolic acids, such as curcumin, kampferol, silymarin, cyanidin-3-glucoside, deosmin, rutin, tartric acid, coumaric acid, and quercetin. This finding is in line with previous reports [58,59]. In addition, this outcome enhances the high antioxidant content of OdLe, as it has been reported that the main biological activity of flavonoids is their antioxidant activity [60]. Specifically, flavonoid antioxidant activity can prevent damage caused by free radicals through the scavenging of reactive oxygen species (ROS), activation of antioxidant enzymes, inhibition of oxidases (e.g., xanthine oxidase [XO], cyclooxygenase [COX], lipoxygenase, and phosphoinositide 3-kinase [PI3K]), and reduction of $\alpha$-tocopheryl radicals [60].

Furthermore, we evaluated cell viability by targeting the ATP intracellular levels of NHDF cells following treatment with OdLe under or without UV irradiation. The ATP intracellular levels of NHDF treated with OdLe were higher compared to the control for all the tested concentrations under or without UVA irradiation. According to previous reports, the induction of intracellular ATP levels is correlated with increased cell viability, the absence of cytotoxicity, and other important cell functions [39,61–63]. Moreover, the induction of ATP intracellular levels even under UVA irradiation suggests the potential role of OdLe in maintaining and recovering the biochemical functions of the cells under UVA irradiation.

In order to understand the mechanism of action of *Origanum dictamnus* L. extract in the cell aging process, we targeted genes that are involved in cell aging in both keratinocytes and fibroblasts under the photoaging process. It is known that during aging, the function of the skin barrier is either altered or disturbed. Thus, we focused on genes that are expressed in the epidermis skin layer and contribute to epidermal skin barrier function, tight junctions, desquamation processes, and epidermal lipid function in an attempt to explore the effect of *Origanum dictamnus* L. extract in the human epidermis. Specifically, the expression of related genes such as *OCLN*, *INHBA*, *UGCG*, *GBA1*, *KLK7*, and *CDSN* was evaluated. The gene *INHBA* encodes a member of the TGF-beta (transforming growth factor-beta) group, which is related to cell proliferation processes [64,65]. Here, the *INHBA* was upregulated post-OdLe addition under UVA irradiation. It has been reported previously that an increase in *INHBA* transcript abundance is associated with enhanced cell proliferation [39]. This induction may emphasize the protective role of OdLe under UVA irradiation, as has been previously highlighted. *OCLN* encodes important tight junction proteins [66,67]. The transcript levels of *OCLN* were upregulated in NHEK cells upon the addition of OdLe with or without UVA irradiation. According to previous reports, induction of *OCLN* is associated with epidermis function [68,69] pointing to the potential contribution of OdLe to the upholding of a healthy epidermis function even under accelerating aging conditions such as photoaging.

In addition, the desquamation process was investigated, as inadequate desquamation causes rough and dry skin, which are characteristics of skin aging. For this purpose, the expression of genes related to the desquamation process, such as *KLK5*, *KLK7*, and *CDSN*, was explored. Specifically, the genes *KLK7* and KLK-5 have an essential role in skin moisturization and in the function of the epidermis barrier as they are involved in the processing of profilaggrin, the precursor of filaggrin, which is essential for keratin structure, skin hydration, and extracellular lipid-processing enzymes in the epidermis [70]. In addition,

the gene *CDSN* stimulates the activity of corneodesmosomes, leading to detachments of corneocytes [71]. The transcript levels of KLK-7 and *CDSN* were upregulated in NHEK cells with the addition of OdLe under UVA irradiation. This outcome suggests the role of OdLe in counteracting the skin's aging process. However, further studies are needed to elucidate this outcome.

Finally, the lipid replenishment system of the skin, focusing on the transcripts of the genes *UGCG* and *GBA1*, was studied. Specifically, the gene *UGCCG* encodes the enzyme that catalyzes the first glycosylation step in the biosynthesis of glycosphingolipids that have an important role in ECM function [72] while the gene GBA1 encodes an acid β-glucosidase that converts (glucosyl) ceramides into ceramides, enhancing the epidermis barrier function [73,74]. The transcript levels of *UGCG* and GBA1 were upregulated in NHEK cells post-OdLe addition under UVA irradiation. According to previous reports, reduction of the transcript levels of *GBA1* as well as UGCG may lead to impaired barrier function associated with skin aging [75,76]. Thus, this outcome deduces a positive effect of OdLe on epidermis barrier function.

We proceed with the investigation of aging-related genes in fibroblasts. Specifically, the expression of the genes *SIRT1*, *SIRT2*, *FOXO3*, *CD44*, *COL3A1*, and *MMP14* was targeted. Sirtuins and FOXO3 are quite well known as cell aging regulators [77] while the genes COL3A1, MMP14, and CD44 are involved in ECM modifications during the cell aging process induced by oxidative stress [78–82]. The expression of the genes *SIRT1*, *SIRT2*, *CD44*, and *MMP14* was upregulated with the addition of OdLe in NHDF cells under UVA irradiation. This observation supports the notion that OdLe stimulates ECM organization while protecting the cells from induced UVA irradiation in a sirtuin-related manner. These findings are in accordance with previous reports [83–85]. Moreover, the expressions of the genes *COL3A1* and *FOXO3* were upregulated with the addition of OdLe, irrespective of the UVA irradiation. Interestingly, this outcome could potentially suggest that OdLe specifically exhibited antiaging activity in a FOXO3- and COL3A1-regulated manner. However, this finding needs to be elucidated further.

The present work describes an attempt to explore the antioxidant profile of the *Origanum dictamnus* L. extract as well as its impact, at a molecular level, upon the cell aging process. Our finding corroborated the strong antioxidant profile of this extract as well as suggesting an interesting role, at a molecular level, for the *Origanum dictamnus* L. extract as it enhances cell proliferation and concomitantly upholds essential biochemical functions of the cells while counteracting the photoaging process. The results of our study could contribute to further understanding of the molecular mechanisms governing anti-aging and stress responses in human dermal cells and may lead to new target molecules for dermocosmetics. The main limitation of our study is that our findings are based on in vitro observations. Nevertheless, future studies will be required to test the hypothesis raised by our data before studies of *Origanum dictamnus* L. efficacy at a clinical level.

## 5. Conclusions

This study explores the antioxidant profile of *Origanum dictamnus* L. extract and its antiaging effects on human fibroblasts and keratinocytes under UVA irradiation.

According to our findings, the OdLe possesses strong antioxidant scavenging activity (DPPH and ABTS assays). Moreover, a high total phenolic content of the OdLe was detected (TPC assay). In addition, the OdLe possesses a Fe(III)-reduction-related activity, which is an important mechanism of phenolic antioxidant action (FRAP assay). This high antioxidant activity of the extract is potentially attributed to the high content of flavonoids and phenolic acids that was detected upon phytochemical screening by LC/MS. The OdLe is rich in curcumin, kampferol, silymarin, cyanidin-3-glucoside, deosmin, rutin, tartric acid, coumaric acid, and quercetin.

Additionally, the gene expression analysis revealed that the OdLe modulates the expression of genes related to the aging process in keratinocytes (KLK7, OCLN, and GBA1) and fibroblasts (SIRT2, FOXO3, and COL3A1) under UVA irradiation. These findings

potentially suggest that the antiaging effect of OdLe in keratinocytes is related mainly to the upholding of a healthy epidermis barrier, while in fibroblasts it is related to the stimulation of ECM organization.

All findings considered, the *Origanum dictamnus* L. extract possesses high antioxidant activity, which is possibly attributed to the high content of flavonoids. From the molecular data, it can be deduced that *Origanum dictamnus* L. extract increases cell viability while counteracting the aging process in different ways in both human fibroblasts and keratinocytes. This compelling synergy of actions makes it a rather promising active agent, exhibiting remarkable potential for enhancing overall skin health and addressing various aging concerns.

**Supplementary Materials:** The following supporting information can be downloaded at: https://www.mdpi.com/article/10.3390/cosmetics10050124/s1, Table S1: Relative mRNA expression of genes, Table S2: Gene name, Accesion No, Kegg pathway, Primer.

**Author Contributions:** Conceptualization, S.L.; methodology, S.L. and M.T.; investigation, S.L., M.T., D.V., A.T., D.A. and D.H.; writing—original draft preparation, S.L., M.T., D.V., A.T., D.A. and D.H.; writing—review and editing, S.L., M.T. and D.H. All authors have read and agreed to the published version of the manuscript.

**Funding:** This research received no funding.

**Institutional Review Board Statement:** Not applicable.

**Informed Consent Statement:** Informed consent was obtained from all subjects involved in the study.

**Data Availability Statement:** Not applicable.

**Conflicts of Interest:** The authors declare no conflict of interest.

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
