# Peer review of "Antioxidant Profile of Origanum dictamnus L. Exhibits Antiaging Properties against UVA Irradiation"

_cosmetics, doi:10.3390/cosmetics10050124_

Round 1

Reviewer 1 Report

The authors have done good work but there are certain issues 

1. Please re-write the abstract, current abstract does not signify the results outcome.

2. In Results section authors did not explain the DPPH, FRAP and ABTS results fully, It just a simple reflection of outputs.

3. In discussion section again discussion about DPPH, FRAP and ABTS completely misining. 

4. In discussion section not even talked about TPC role.

5. From Supplementary material files, DPPH, FRAP and ABTS graphs did not describe the complete sense, readers or future researchers will not get anything from those graphs. 

6. Please re-write conclusion also to justify the outcomes. 

Author Response

The authors have done good work but there are certain issues

Answer: We thank the Reviewer for their critical reading of our MS and for the time they have taken to provide this very helpful feedback. We have taken all the reviewer’s comments into consideration and have modified our MS accordingly. We are convinced the changes have contributed to improve our MS. To facilitate identification of the changes we have used red typeface in the revised MS.

  1. Please re-write the abstract, current abstract does not signify the results outcome.

Answer: We appreciate the Reviewer’s candid comments in relation to the lack of clarity of the abstract. We have now rewritten the abstract of the MS to signify the outcome of our study (ln 10 – 26).

  1. In Results section authors did not explain the DPPH, FRAP and ABTS results fully, It just a simple reflection of outputs.

Answer: We appreciate the Reviewer’s comments in the results section in relation to the DPPH, FRAP and ABTS assays. We have now rewritten this section of the MS to explain better the outcomes of these assays (ln 194-211).

  1. In discussion section again discussion about DPPH, FRAP and ABTS completely misining.
  2. In discussion section not even talked about TPC role.

Answer: We agree with the Reviewer’s candid comments in relation to the discussion part of the MS. We have now discussed our findings based on enzymatic assay. Please check ln 284– 297. We have also rewritten most of the discussion part of the MS.

  1. From Supplementary material files, DPPH, FRAP and ABTS graphs did not describe the complete sense, readers or future researchers will not get anything from those graphs.

Answer: We appreciate the Reviewer’s critical comment on the supplementary materials. We have now eliminated these graphs from the supplementary materials while we enhanced the methodology of these assay in the materials and methods section of the MS. Please check ln.84-141 of the MS.

  1. Please re-write conclusion also to justify the outcomes.

Answer: We agree with the Reviewer’s candid comment in relation to the conclusion part of the MS. We have now rewritten the conclusions of the MS to signify underlining our outcomes. Please check ln 387-408 of the manuscript.

Reviewer 2 Report

This is interesting manuscript on a well conducted study in the field plant antioxidant in skin aging. I believe the study adds to the body of literature in cosmetology and plant science field. However, I do have several concerns major concerns which should be improved prior to publication:

  1. English language must be improved 
  2. Abstract should be more informative, please describe your methodology and results, not just theory 
  3. Line 28 - is diabetes really number one example for degenerative disease associated with aging? Also, you are missing spaces before references 
  4. You defined ROS twice
  5. Line 67 - if it is approved for traditional use that means it does not have enough evidence for well established use, please explain that in this paragraph
  6. You should describe your methods (each one) because reader should know which reagents have been used (manufacturer)
  7. English is really poor in the results section, this can not be published in this form
  8. Figures are hard to read, please improve their quality 

Really poor 

Author Response

This is interesting manuscript on a well conducted study in the field plant antioxidant in skin aging. I believe the study adds to the body of literature in cosmetology and plant science field. However, I do have several concerns major concerns which should be improved prior to publication:

Answer: We appreciate the Reviewer’s critical consideration of our MS and their positive comments about our work. We thank the reviewer for the time they have taken to carefully read and critically evaluate our MS. We have modified our MS taking into consideration all the reviewer’s comments. We are convinced the changes have contributed to improve our MS. To facilitate identification of the changes we have used red typeface in the revised MS.

English language must be improved

Abstract should be more informative, please describe your methodology and results, not just theory

Answer: We appreciate the Reviewer’s candid recommendations of our MS.

We have is now improved the English level of our MS as it has been checked by a native English-speaking colleague. Moreover, we have improved the abstract according to Reviewer’s recommendations. Please check ln. ln 10 – 26.

Line 28 - is diabetes really number one example for degenerative disease associated with aging? Also, you are missing spaces before references

You defined ROS twice

Line 67 - if it is approved for traditional use that means it does not have enough evidence for well established use, please explain that in this paragraph

Answer: We appreciate the Reviewer’s critical comments on the introduction section of the MS. We have now rewritten these parts eliminating any vague wording. Please check ln.31, ln.41-52 and ln.70-71 of the MS.

You should describe your methods (each one) because reader should know which reagents have been used (manufacturer)

English is really poor in the results section, this can not be published in this form

Figures are hard to read, please improve their quality

Answer: We appreciate the Reviewer’s comment on the “materials and method” and “results” section of the MS. We have now rewritten these sections of the MS and improve the quality of the figures according to journal’s and reviewer’s recommendations. Please check ln.84-141, ln. 194-264 as well as the figures of the MS.

Reviewer 3 Report

·       - The manuscript seems to have some problems that need to be solved by the authors.

·        - In the abstract, the sentence "The aim of this study was to assess antioxidant profile of Origanum dictamnus L. extract as well as its antiaging effects on 2D cultures of fibroblasts as well as keratinocytes under UVA irradiation so as to explore the impact of Origanum dictamnus L. extract in cell aging process" could be refined for readability.

·        - The extraction should be repeated to be sure that all the active principles were extracted.

·        - The authors did not mention about the identification of the plant and about the voucher specimen that should be deposited for further experiments.

·        - What positive controls were used in the antioxidant assays?

·        - Every study has limitations, and it would be valuable to briefly address potential limitations of the study's design or methodology. This helps readers understand the scope and boundaries of the research.

- Including a sentence or two about the potential implications of the findings for the skincare industry or further research directions could enhance the research impact.

The English requires revision

Author Response

- The manuscript seems to have some problems that need to be solved by the authors.

Answer: We thank the Reviewer for their critical reading of our MS and for the time they have taken to provide this very helpful feedback. We have taken all the reviewer’s comments into consideration and have modified our MS accordingly. We are convinced the changes have contributed to improve our MS. To facilitate identification of the changes we have used red typeface in the revised MS.

  • - In the abstract, the sentence "The aim of this study was to assess antioxidant profile of Origanum dictamnus L. extract as well as its antiaging effects on 2D cultures of fibroblasts as well as keratinocytes under UVA irradiation so as to explore the impact of Origanum dictamnus L. extract in cell aging process" could be refined for readability.

Answer: We appreciate the Reviewer’s candid comments in relation to the lack of clarity of the abstract. We have now rewritten the abstract of the MS to signify the outcome of our study (ln 10 – 26).

  • - The extraction should be repeated to be sure that all the active principles were extracted.

  • - The authors did not mention about the identification of the plant and about the voucher specimen that should be deposited for further experiments.

Answer: We appreciate the Reviewer’s critical comments in the methodology of our MS. We would like to underline that the identification of the plant wasn’t on the scopus of this article as it was already identified by our provider. We have now rewritten these parts of the MS according to reviewer’s recommendations. Please check ln. 82-83.

  • - What positive controls were used in the antioxidant assays?

Answer: We appreciate the Reviewer’s candid comments in relation to the methodology of the antioxidant assays. We have now rewritten this section of the MS. Please check ln.84-141 of the MS.

  • - Every study has limitations, and it would be valuable to briefly address potential limitations of the study's design or methodology. This helps readers understand the scope and boundaries of the research.

 Answer: We thank the Reviewer’s for this suggestion. We have now briefly addressed the main limitation of the study in the discussion section of our MS. Please check ln.  375-385 of the MS.

- Including a sentence or two about the potential implications of the findings for the skincare industry or further research directions could enhance the research impact.

Answer: We thank the Reviewer’s for this suggestion. We have now rewritten the discussion and the conclusion part of our MS including some sentences of potential implications in skincare industry as well as further future direction of our research in the discussion part. Please check ln. 273-385 of the MS

Round 2

Reviewer 1 Report

Good Work

Reviewer 2 Report

I believe it is now appropriate for publication 

Reviewer 3 Report

The authors responded properly to my inquiries. I consider the article proper to be published.

The authors improved the quality of English